# Evaluation of the NG-Test CARBA 5 for Rapid Detection of Carbapenemases in Clinical Isolates of *Klebsiella pneumoniae*

**DOI:** 10.3390/antibiotics14100989

**Published:** 2025-10-02

**Authors:** Bojan Rakonjac, Momčilo Djurić, Danijela Djurić-Petković, Jelena Dabić, Marko Simonović, Marija Milić, Aleksandra Arsović

**Affiliations:** 1Institute of Microbiology, Military Medical Academy, 11040 Belgrade, Serbia; bonatejo@gmail.com (B.R.); momdjuric@gmail.com (M.D.); danijeladjuricpetkovic@gmail.com (D.D.-P.); jelenadabic11@gmail.com (J.D.); 2Medical Faculty of the Military Medical Academy, University of Defence, Serbia Crnotravska 17, 11000 Belgrade, Serbia; 3Institute of Epidemiology, Military Medical Academy, 11040 Belgrade, Serbia; markosimonovic84@gmail.com; 4Department of Epidemiology, Faculty of Medicine, University of Pristina, 38220 Kosovska Mitrovica, Serbia; marijamilic85@gmail.com

**Keywords:** *Klebsiella pneumoniae*, carbapenemase, NG-Test CARBA 5, PCR, antimicrobial resistance, rapid diagnostics

## Abstract

**Background:** Carbapenem-resistant *Klebsiella pneumoniae* (CRKp) is a critical global health threat due to its multidrug resistance, primarily driven by carbapenemase production. Rapid and accurate detection of carbapenemases is essential for effective treatment and infection control. This study evaluates the validity of the NG-Test CARBA 5, a rapid immunochromatographic assay, for detecting five major carbapenemases (KPC, NDM, VIM, IMP, OXA-48-like) in clinical CRKp isolates. **Methods:** Clinical isolates of CRKp were collected from various clinical specimens at the Military Medical Academy in Belgrade, Serbia, between January 2023 and October 2024. Detection of carbapenemases was performed using NG-Test CARBA 5, while PCR served as the reference method. Diagnostic performance was assessed by calculating sensitivity, specificity, and Cohen’s kappa coefficient. **Results:** Among 312 isolates, OXA-48-like was the most prevalent carbapenemase. NG-Test CARBA 5 showed high sensitivity (98.7%) and specificity (100%) overall, with excellent agreement for NDM (κ = 0.947), OXA-48-like (κ = 0.957), and KPC (κ = 0.978). However, it failed to detect VIM in five PCR-positive isolates, suggesting potential limitations. **Conclusions:** NG-Test CARBA 5 is a rapid and reliable tool for detecting major carbapenemases in CRKp, though its performance for VIM detection requires further investigation. This assay has the potential to improve clinical diagnostics and strengthen infection control in settings with high antimicrobial resistance.

## 1. Introduction

Antimicrobial resistance is a major global health threat of the 21st century [1]. The increasing prevalence of drug-resistant bacteria impacts and complicates the treatment of infections, increases morbidity and mortality, prolongs hospital stays, and increases healthcare costs. A report by the Centers for Disease Control and Prevention (CDC) in 2019 stated that AMR is directly responsible for at least 1.27 million deaths worldwide and is associated with about 5 million deaths [1].

Carbapenems are often considered the last line of defense for serious infections caused by Gram-negative bacteria. However, emerging resistance to carbapenems severely limits treatment options. The major contributors to multi-drug resistance (MDR) are carbapenem-resistant Enterobacterales (CRE), which are listed as one of the critical priority groups posing the highest threat to public health. Infections with these pathogens are highly transmissible, difficult to prevent, and associated with an increased death risk [1,2]. Most infections caused by CRE are seen in hospital settings in patients on mechanical ventilation, using urinary catheters, intravenous catheters, on long-term antibiotic treatment, as well as immunocompromised patients [2,3].

As a member of CRE, carbapenem-resistant *Klebsiella pneumoniae* (CRKp) belongs to the critical priority pathogens as classified by the World Health Organization (WHO) [1]. It is a major opportunistic pathogen and a common cause of hospital-acquired infections. The underlying cause is the interplay between diverse virulence determinants and the broad acquisition of antimicrobial resistance mechanisms [4].

A global meta-analysis by Lin et al. (2024) estimated that approximately 28.7% of hospital-acquired *K. pneumoniae* infections were resistant to carbapenems, with high inter-country heterogeneity [5]. Recent surveillance data reported a 57.5% increase in the incidence of bloodstream infections caused by CRKp, with higher AMR percentages in southern and eastern Europe compared to northern Europe [6,7].

CRKp has emerged as a major public health concern in the Balkan region, with Serbia being one of the most affected countries [8,9]. One study showed a rapid increase in CRKp prevalence in blood cultures, with a rise from 11.76% in 2020 to 72.94% in 2022 [10]. Also, a shift in dominant carbapenemase types is noticed. Earlier studies showed dominance of NDM type, followed by OXA-48 and KPC [11]. Later genomic studies reveal OXA-48 as the dominant carbapenemase in CRKp isolates, usually associated with high-risk clone ST101 [12,13].

The capacity of *K. pneumoniae* to acquire, accumulate, and disseminate resistance determinants renders it a critical concern. The major and most common mechanism of carbapenem-resistance in CRKp is the production of carbapenemases [14]. The major carbapenemases responsible for carbapenem resistance in *K. pneumoniae* are: Ambler class A K. pneumoniae carbapenemase (KPC), Ambler class B metal-lo-β-lactamases (MBL), including New Delhi metallo-β-lactamase (NDM), Verona integron-encoded metallo-β-lactamase (VIM), imipenemase (IMP), and Ambler class D oxacillinase β-lactamase-48 (OXA-48-like) [15].

Since CPKp can lead to life-threatening infections [16,17], accurate and timely detection of carbapenemase production is essential in routine microbiological diagnostics, as it directly influences both clinical decision-making and infection control strategies [16,17,18]. Additionally, because the susceptibility of isolates containing different carbapenemases may differ, in order to properly direct antibiotic therapy, it is important to accurately identify serine versus MBL carbapenemases. For example, the β-lactam/β-lactamase inhibitor ceftazidime/avibactam combination works well against class A and class D carbapenemases, but not against class B metalloenzymes [19], which have the ability to hydrolyze the new β-lactamase inhibitors [20]. The NG-Test CARBA 5 (NG Biotech, Guipry, France) is an immunochromatographic assay that utilizes monoclonal antibodies to detect the five major carbapenemases (KPC, NDM, VIM, IMP, and OXA-48-like) directly from bacterial colonies, offering results within 15 min through a simple procedure and easy-to-interpret test strips [21,22,23,24].

Given the limited number of studies evaluating the performance of the NG-Test CARBA 5 under real laboratory conditions, particularly those testing a variety of real and mixed clinical specimens (combining both real clinical and spiked cultures), rather than artificially inoculated media [16,23,25,26,27], our study aimed to evaluate the performance and clinical usefulness of the NG-Test CARBA 5 for routine laboratory use in detecting and distinguishing the five major carbapenemases among clinical isolates of CPKp. Conventional PCR was used as the reference standard to evaluate the test accuracy within a typical diagnostic workflow.

## 2. Results

### 2.1. Distribution of PCR-Identified Genes and Enzymes Detected by NG-Test CARBA 5

A total of 312 isolates were analyzed. Blood cultures were the most frequent specimen source (54.8%), followed by aspirates (21.5%) and urine cultures (10.9%). Confirmatory PCR analysis revealed that the most commonly detected gene was *bla*_OXA-48-like_ (67.3%), followed by *bla*_NDM_ (24.7%) and *bla*_KPC_ (18.3%). *bla*_VIM_ was detected in 1.6% of isolates, while *bla*_IMP_ was not detected. Correspondingly, the NG-Test CARBA 5 identified OXA-48-like in 65.4%, NDM in 22.8%, and KPC in 17.6% of samples, with no detection of VIM or IMP. The summarized concordance in gene detection demonstrated that the majority of isolates harbored a single gene (87.2% by PCR; 90.4% by NG-Test CARBA 5), whereas dual gene presence was observed in a smaller proportion (11.9% by PCR; 7.7% by NG-Test CARBA 5). PCR identified the simultaneous presence of three genes in only one isolate (0.3%) (Table 1).

### 2.2. Comparison of NG-Test CARBA 5 Performance to PCR as the Reference Method for Carbapenemase Gene Detection

Comparison of the diagnostic performance of the NG-Test CARBA 5 with PCR as the reference method for detecting individual carbapenemase genes showed that the NG-Test CARBA 5 demonstrated high sensitivity and perfect specificity for NDM (92.2% sensitivity, 100% specificity), OXA-48-like (97.1%, 100%), and KPC (96.5%, 100%) (Table 2). For VIM and IMP, NG-Test CARBA 5 showed no positive results, resulting in undefined sensitivity due to the absence of positive PCR-confirmed cases for *bla*_IMP_, and low detection for *bla*_VIM_. The overall sensitivity and specificity of NG-Test CARBA 5 in comparison to PCR for all genes combined were 98.7% and 100%, respectively. The positive predictive value was 100% for all targets, while the negative predictive value ranged from 94.4% to 99.2%, except for the total PCR-negative cases, where it was 66.7% (Table 2).

### 2.3. Agreement Between NG-Test CARBA 5 and PCR

The agreement between NG-Test CARBA 5 and PCR results was evaluated using Cohen’s kappa coefficient. There was excellent agreement for the detection of NDM (κ = 0.947), OXA-48-like (κ = 0.957), and KPC (κ = 0.978), all with statistically significant *p*-values (<0.001). No agreement was observed for VIM (κ = 0.000, *p* = 1.000), and kappa values for IMP could not be calculated due to the absence of positive cases. The overall agreement between NG-Test CARBA 5 and PCR across all gene targets was moderate (κ = 0.495, *p* < 0.001) (Table 3).

## 3. Discussion

The emergence and spread of CRKp represent a significant public health concern due to the rapid development of resistance, especially in hospital settings. Carbapenem resistance, primarily mediated by the production of carbapenemases, highlights the urgent need for rapid diagnostics and appropriate therapeutic strategies [28,29].

Many studies have indicated carbapenemase-producing Enterobacterales (CPE) as the predominant type of carbapenem-resistant Enterobacterales (CRE) with prevalence rates ranging from 77.3% to 91.3% [30,31,32]. Also, in our study, 99.36% (310/312) of clinical CRKp produced carbapenemases, indicating that the main mechanism of resistance among clinical CRKp isolates is carbapenemase production.

Of the 310 CPKp isolates, OXA-48-like carbapenemase was the most prevalent, which is consistent with a study of *K. pneumoniae* in community settings in Belgrade, Serbia [8]. Recent data from Serbia indicate that *bla*_OXA48-like_ is the most prevalent carbapenemase gene among hospital *K. pneumoniae* isolates. In a nationwide survey, MDR *K. pneumoniae* was detected in 8.3% of hospital isolates, often co-occurring with *bla*_NDM_ [11]. Also, subsequent studies have shown an increasing dominance of *bla*_OXA48-like_ from different clinical samples [10,13]. Genomic analyses confirmed the widespread distribution of *bla*_OXA48-like_ on novel plasmids among ST101 strains [12]. Similarly, in a cohort of preterm neonates at hospital discharge, *bla*_OXA48-like_ was detected in 47.7% of carbapenem-resistant K. pneumoniae colonizing isolates [33]. In contrast, *bla*_VIM_ and *bla*_IMP_ have been consistently absent or rarely detected in clinical isolates across studies, suggesting limited circulation of these metallo-β-lactamase genes in Serbia to date [34].

CPKp is often resistant to multiple antibiotics and has limited treatment options. In terms of treatment, the choice of drugs to treat CRE infection depends on specific carbapenemases [35]. For KPC and OXA-48-like enzymes, ceftazidime-avibactam can be the preferred agent [36]. Unfortunately, ceftazidime-avibactam is not active on MBL producers, and other options must be investigated, such as combining the remaining active antibiotics, (e.g., colistin, tigecycline, aminoglycosides, and/or fosfomycin) [37], or aztreonam in combination with ceftazidime-avibactam [38,39]. Consequently, it is crucial to detect and identify carbapenemases as soon as possible, so that the physician can quickly apply or change antibiotic therapy and implement appropriate infection control measures. NG-Test CARBA 5 is an effective, quick, and practical diagnostic technique that may aid in streamlining the intricate regular workflow for carbapenemase detection.

In our study, by comparing NG-Test CARBA 5 to the reference PCR method, the overall sensitivity and specificity were 98.7% and 100%, respectively. Our results show excellent performance of NG-Test CARBA 5 for detecting major variants of carbapenemases (OXA-48-like and KPC) among CPKp (sensitivity and specificity > 95%). However, it is important to note that the NG-Test CARBA 5 gave a false negative result for four CPKp isolates and missed four single NDM-producing strains. Similarly, in our study, Baer et al. [40] correctly identified 89.5% of the strains tested, while 10.5% of the isolates showed a false negative result. They detected 80% of NDM, 83.3% of VIM, 87.5% of OXA-48-like, and 100% of KPC producers. However, the test failed to recognize a single IMP-producing pathogen. Contrary to our results, Pruss et al. [41] found 100% of strains producing carbapenemases of the NDM, VIM, KPC, and OXA-48-like types, and negative results were obtained for strains not producing carbapenemases.

In addition to the four NDM isolates missed, the other missed carbapenemases in our study are from isolates coproducing two or three types of carbapenemases, including NDM + VIM (n = 2), OXA-48-like + VIM (n = 2), KPC + VIM (n = 1), NDM + OXA-48-like (n = 5), OXA + KPC (n = 4), and NDM + OXA-48-like + KPC (n = 1). In contrast, twenty-four NDM + OXA-48-like coproducers were accurately identified by NG-Test CARBA 5, suggesting that strains producing multiple carbapenemases are very common in our hospital environment. Also, Liu et al. [42] showed that the NG test missed eight NDM, four OXA-48-like, and one IMP blood isolates, especially with co-carriage of carbapenemases compared with the Xpert CARBA-R assay. On the other hand, in a study from Guangdong, China, seven isolates coproduced two types of carbapenemases, including NDM + IMP (n = 5), KPC + NDM (n = 1), and OXA-48-like + IMP (n = 1), which were accurately identified by NG-Test CARBA 5 [43].

Most importantly, in our study, the performance of the NG-Test CARBA 5 may also be affected by variants of carbapenemases, especially false-negative VIM isolates. Five CPKp isolates harboring the *bla*_VIM_ gene were detected by PCR analysis but were missed by the NG-Test CARBA 5 assay. There are several possible explanations for this. Firstly, we speculated on possibility that bacterial colonies utilized to prepare immunochromatographic tests <,and that are used for DNA isolation, may contain clonally related isolates with different resistotypes, that were not recognized, and which could influence our data analysis Secondly, there may be a false-negative result due to the nature of the NG-Test CARBA 5 method (a multiplex immunochromatographic assay, which is based on the detection of expressed proteins by antibodies) if the carbapenemases are not well expressed or if they contain amino acid substitutions at the primary epitope of the antibody [44,45]. Finally, a PCR may yield false-positive results due to mutations in the structural gene or in its promoter, which impair its expression [45].

A notable limitation of our study concerns the detection of the VIM-producing *K. pneumoniae* isolates. While the NG-Test CARBA 5 assay did not detect VIM carbapenemase, PCR analysis identified five *K. pneumoniae* isolates carrying the *bla*_VIM_ gene. Since we were unable to perform gene sequencing analysis to confirm PCR findings, the possibility of false-positive results, non-specific amplification, or the presence of a non-functional or partial gene sequence cannot be ruled out. Thus, we cannot state with certainty whether the NG-Test CARBA 5 failed to detect VIM-positive isolates. In order to fully assess the sensitivity and specificity of the NG-Test CARBA 5 in detecting VIM carbapenemase, further studies that incorporate sequencing analysis are necessary to definitively confirm the presence of the *bla*_VIM_ gene and validate the observed discrepancy between the PCR and NG-Test CARBA 5 results. Saito et al. [27] also highlighted in their study that the detection of VIM carbapenemases, particularly those with low expression levels, remains a challenge for NG-Test CARBA 5.

Our study confirms that NG-Test CARBA 5 is highly accurate in detecting the most common carbapenemases among clinical CPKp isolates, with an overall sensitivity of 98.7% and specificity of 100% when compared to the PCR method. These results largely align with studies that used real clinical samples, such as the research by Boutal et al. [23], which demonstrated a high sensitivity of 97.7% and specificity of 96.1% in clinical blood cultures. Additionally, our findings are consistent with studies that used a mixed culture, combining real clinical and spiked cultures, where sensitivity ranged from 94 to 98% and specificity from 97 to 99% for NG-Test CARBA 5 [16,25,26,27]. Qin et al. [46], in their systematic review of various studies evaluating the validity of the NG-Test CARBA 5, highlighted the importance of using real clinical samples in diagnostic assessments. Their review indicated that while spiked cultures are valuable for controlled testing, they do not fully replicate the complexities of patient-derived isolates. These complexities, such as mixed bacterial populations, host factors, and microbial interactions, can significantly influence the performance and accuracy of diagnostic tests like NG-Test CARBA 5. The review also noted a difference in sensitivity and specificity between real clinical samples and spiked cultures. Specifically, studies using real clinical samples often reported higher sensitivity and specificity compared to those using spiked cultures, where test performance could be affected by the artificial nature of the samples. Therefore, our data highlight the advantages of using real clinical samples in the evaluation of diagnostic tests. Our study strongly supports the use of real clinical samples in assessing NG-Test CARBA 5, as it provides a more realistic view of its diagnostic accuracy and reliability. Also, NG-Test CARBA 5 proved to be accurate and reliable test in detecting the most common carbapenemases among clinical CPKp isolates, with high sensitivity and specificity.

## 4. Methods

### 4.1. Isolates

Between January 2023 and October 2024, 312 strains of CRKp isolated from various clinical samples, including urine culture, blood culture, bronchoalveolar lavage (BAL), aspirate, kiss swab, and liquor originating from different wards in the Military Medical Academy, Serbia, were included in the study. All samples were cultured on blood agar with 5% sheep blood (bioMérieux, Marcy-l’Étoile, France) and incubated overnight at 37 °C in an aerobic atmosphere. After incubation, colonies consistent with *Klebsiella* morphology were identified to the species level by the MALDI-TOF MS method (Vitek MS, bioMérieux, Marcy-l’Étoile, France).

Susceptibility to carbapenem antibiotics was performed during the routine antibiotic testing diagnostic procedure by using the disk diffusion method according to Kirby–Bauer [17]. Resistance to carbapenem antibiotics was confirmed with meropenem (10 µg) and ertapenem (10 µg) disks (BioRad, Marnes-la-Coquette, France) according to the EUCAST methodology. For some species, the minimal inhibitory concentration (MIC) of meropenem and ertapenem was also determined by quantitative assay for determining MIC of antimicrobial agents, MIC Test Strip (Liofilchem, Roseto degli Abruzzi, Italy).

According to EUCAST guidelines, *K. pneumoniae* isolates that demonstrated an inhibition zone for ertapenem and/or meropenem < 25 mm using the disk diffusion method and <2 mg/L by the gradient test method were chosen for additional examination [47].

### 4.2. NG-Test CARBA 5 Assay

The NG-Test CARBA 5 Assay, for the five major carbapenemases KPC, IMP, NDM, VIM, and OXA-48-like, was carried out in accordance with the manufacturer’s instructions (NG Biotech, Guipry, France). In short, 100 µL of the mixture was poured onto the CARBA-5 cassette after a 1 µL loopful of bacteria and five drops of extraction buffer were combined. The mixture should be vortexed and then allowed to sit at room temperature (20 to 25 °C) for 15 min. After 15 min of room temperature incubation, the results were evaluated [13].

### 4.3. DNA Isolation and Detection of Carbapenemase Genes by PCR

DNA was extracted using the boiling method. A couple of colonies from an agar plate were transferred to Luria–Bertani broth (LB) and incubated overnight. From the overnight culture, 1500 µL of bacteria were centrifuged for 2 min at 12,000 rpm. The pellet was resuspended in 300 µL of sterile, distilled water, boiled for 10 min, and transferred to −20 °C for 10 min. After that, tubes were centrifuged for 2 min at 12,000 rpm, and the supernatant containing the DNA was transferred to a new tube and used for PCR or kept at −20 °C. Detection of carbapenemase genes was carried out using primers in Table 4. The genes *bla*_NDM_, *bla*_KPC_, *bla*_OXA-48_, *bla*_VIM,_ and *bla*_IMP_ were detected by multiplex PCR reaction. The PCR conditions were: initial denaturation at 94 °C for 5 min, followed by 30 cycles of denaturation at 94 °C for 45 s, annealing at 59 °C for 60 s, extension at 72 °C for 60 s, and the final extension at 72 °C for 10 min. The PCR products were analyzed by electrophoresis in 2% agarose gel, stained with ethidium bromide, and visualized using a UV transilluminator [48,49,50,51].

### 4.4. Statistical Analysis

The NG-test CARBA 5 sensitivity and specificity were computed using a 95% confidence interval (CI). Each method’s agreement with the PCR as the gold standard was assessed using the Kappa index (κ index). SPSS Statistics v15.0 was used to analyze the data (IBM, Chicago, IL, USA).

## 5. Conclusions

Based on our results and those of previous studies, it can be concluded that NG-Test CARBA 5 is a rapid, reliable, and easy-to-implement in routine workflow tool for detecting major carbapenemases in CRKp. The assay could be very useful in assisting with clinical diagnostics, optimal therapy choice, and infection control in high-resistance clinical settings. It showed high accuracy with good sensitivity and specificity for the detection of clinically significant carbapenemases NDM, OXA-48-like, and KPC, though its performance for VIM detection requires further investigation.

## Figures and Tables

**Table 1 antibiotics-14-00989-t001:** Detection results of carbapenemase genes by PCR and carbapenemses by NG-Test CARBA 5.

Categories		Number of Isolates (Percentage %)
Specimen	Urine culture	34 (10.9)
	Blood culture	171 (54.8)
	BAL	13 (4.2)
	Aspirate	67 (21.5)
	Kiss swab	25 (8.0)
	Liquor	2 (0.6)
PCR-NDM	Negative	235 (75.3)
	Positive	77 (24.7)
PCR-OXA48	Negative	102 (32.7)
	Positive	210 (67.3)
PCR-KPC	Negative	255 (81.7)
	Positive	57 (18.3)
PCR-VIM	Negative	307 (98.4)
	Positive	5 (1.6)
PCR-IMP	Negative	312 (100.0)
	Positive	0 (0.0)
NG-Test CARBA 5/NDM	Negative	241 (77.2)
	Positive	71 (22.8)
NG-Test CARBA 5/OXA48	Negative	108 (34.6)
	Positive	204 (65.4)
NG-Test CARBA 5/KPC	Negative	257 (82.4)
	Positive	55 (17.6)
NG-Test CARBA 5/VIM	Negative	312 (100.0)
	Positive	0 (0.0)
NG-Test CARBA 5/IMP	Negative	312 (100.0)
	Positive	0 (0.0)
All PCR	Negative	2 (0.6)
	One gene detected	272 (87.2)
	Two genes detected	37 (11.9)
	Three genes detected	1 (0.3)
All NG-Test Carba5	Negative	6 (1.9)
	Single carbapenemase detected	282 (90.4)
	Double carbapenemases detected	24 (7.7)

**Table 2 antibiotics-14-00989-t002:** Diagnostic performance of NG-Test CARBA 5 compared to PCR as the reference method for carbapenemase gene detection.

Results by Reference Methods	NG CARBA 5							
	TP	FP	TN	FN	Sensitivity (%)	Specificity (%)	Positive Predictive Value (%)	Negative Predictive Value (%)
PCR-NDM	71	0	235	6	92.2	100.0	100.0	97.5
PCR-OXA48	204	0	102	6	97.1	100.0	100.0	94.4
PCR-KPC	55	0	255	2	96.5	100.0	100.0	99.2
PCR-VIM	0	0	307	5	/	100.0	/	98.4
PCR-IMP	0	0	312	0	/	100.0	/	100.0
ALL PCR	306	0	2	4	98.7	100.0	100.0	66.7

TP—true-positive, FP—false-positive, TN—true-negative, and FN—false-negative.

**Table 3 antibiotics-14-00989-t003:** Agreement between NG-Test CARBA 5 and PCR results for detection of carbapenemase (Cohen’s kappa analysis).

PCR vs. NG-Test CARBA 5	Kappa	*p*
PCR-*bla*_NDM_ vs. NG CARBA 5 NDM	0.947	<0.001
PCR-*bla*_OXA48-like_ vs. NG CARBA 5 OXA48-like	0.957	<0.001
PCR-*bla*_KPC_ vs. NG CARBA 5 KPC	0.978	<0.001
PCR-*bla*_VIM_ vs. NG CARBA 5 VIM	0.000	1.000
PCR-*bla*_IMP_ vs. NG CARBA 5 IMP	/	/
ALL blaPCR vs. all NG CARBA 5	0.495	<0.001

**Table 4 antibiotics-14-00989-t004:** Primers used for carbapenemases gene detection.

Primer	Sequence 5′–3′	Amplicon Size (bp)	Reference
*bla*_KPC_ Fw	ATGTCACTGTATCGCCGTCT	893	[48]
*bla*_KPC_ Rw	TTTTCAGAGCCTTACTGCCC		
*bla*_VIM_ Fw	GATGGTGTTTGGTCGCATA	390	[49]
*bla*_VIM_ Rw	CGAATGCGCAGCACCAG		
*bla*_NDM_ Fw	GGGCAGTCGCTTCCAACGGT	475	[50]
*bla*_NDM_ Rw	GTAGTGCTCAGTGTCGGCAT		
*bla*_IMP_ Fw	GGAATAGAGTGGCTTAATTCTC	188	[49]
*bla*_IMP_ Rw	CCAAACCACTACGTTATCT		
*bla*_OXA-48-like_ Fw	TTGGTGGCATCGATTATCGG	744	[51]
*bla*_OXA-48-like_ Rw	GAGCACTTCTTTTGTGATGGC		

## Data Availability

The data presented in this study are available in this article.

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
