# Peer review of "Evaluation of the NG-Test CARBA 5 for Rapid Detection of Carbapenemases in Clinical Isolates of Klebsiella pneumoniae"

_antibiotics, 2025, doi:10.3390/antibiotics14100989_

Round 1
Reviewer 1 Report
Comments and Suggestions for Authors
Carbapenem-resistant Klebsiella pneumoniae is a critical global health threat due to its multidrug resistance, mainly due to carbapenemase production. Τhe authors evaluated the performance of the NG Test Carba 5 in detecting and distinguishing the five major carbapenemases among carbapenem-resistant Klebsiella pneumoniae clinical isolates. It is essential that the evaluation was done under real laboratory conditions using patient-derived isolates.
I enjoyed reading a well-written paper, and I would like to raise only two points
Line 146: (Table 3) change to (Table 4)
Lines 158 and 159: Enterobacteriaceae change to Enterobacterales
Author Response
Comment 1:
Line 146: (Table 3) change to (Table 4)
Answer 1: Thank you for pointing out this mistake. We have corrected it.
Comment 2:
Lines 158 and 159: Enterobacteriaceae change to Enterobacterales
Answer 2: Thank you for noticing our mistake. We have corrected it accordingly.
Reviewer 2 Report
Comments and Suggestions for Authors
The identification of carbapenemases is a major problem in the management of drug-resistant Klebsiella pneumoniae infections. I believe that the NG-Test Carba 5 is a useful test in clinical practice, which can significantly contribute to the rapid detection of the type of carbapenemases, bringing a major benefit to clinical practice. Although there are problems in detecting VIM carbapenemase, this method brings major benefits, given the rapidity, specificity, and sensitivity of the test for the other known types of carbapenemases.
Author Response
We would like to thank the reviewer for the time and effort invested in revising and evaluating our manuscript.
Reviewer 3 Report
Comments and Suggestions for Authors
- Introduction, write ‘CPKp can lead to life-threatening infections (5, 6). Accurate and timely detection of carbapenemase…………………….’
- Write ‘The mixture should be vortexed and then allowed to sit at room temperature 95
(20 to 25 °C) for 15 minutes. - Please correct KPC in this sentence : The genes blaNDM, blaKP, blaOXA-48, and blaVIM were detected by multiplex PCR reaction’.
- Write ‘The PCR conditions were: initial denaturation at 94 °C for 5 min, followed
by 30 cycles of denaturation at 94 °C for 45 s, annealing at 59 °C for 60 s, and extension at 72 °C for 60 s, and the final extension at 72 °C for 10 min.’ - Please correct ‘blaNDM Rw’ an blaOXA-48-like in table 1.
- You can delete the footnote under table1; really it is not necessary it exist in the manuscript and are well known enzymes (KPC – Klebsiella pneumoniae carbapenemase; VIM – Verona integron-encoded metallo--lactamase; NDM – New Delhi metallo-β-lactamase; IMP – imipenemase; OXA-48-like – Oxacillinase-48-like).
- Table 2, you write frequency, this is not a frequency this is a number. So I advide to write simply: Number of isolates (Percentage %), so it is a unique column, for example for the line urine culture in its front just one column containing: 34 (1.9), for PCR OXA-48 /Negative/ 102 (32.7), ......the remaining lines also........... I hope that you understand me.
- Please below table 3, add abbreviation of TP, FP, TN, FN
- In this sentence ‘The overall agreement between NG-Test Carba 5 and 145
PCR across all gene targets was moderate (κ = 0.495, p < 0.001) (Table 3).’ I believe you mean table 4 not table 3. - You wrote ‘There are several possible explanations for this. Firstly we speculated on possibility that bacterial colonies utilized to prepare immunochromatographic tests and DNA isolation may contain clones that were not recognized, and which could influence our data analysis’. THIS hypothesis is not clear, as written it is not correct scientifically, I advise that you read this and see if it correspond really to what you want say. Clones not recognized?!
- Please verify this hypothesis it seems not correct or not clear ‘Finally, a PCR may yield false-positive results if a mutation alters the carbapenemase activity but not the primer binding sites or gene (40).’ You mean that the gene exists but not expressed owing to mutation(s) in the structural gene or in its promoter.
Author Response
Comment1:
Introduction, write ‘CPKp can lead to life-threatening infections (5, 6). Accurate and timely detection of carbapenemase…………………….’
Answer 1: As per your suggestion, we have corrected the sentence.
Comment 2:
Write ‘The mixture should be vortexed and then allowed to sit at room temperature 95
(20 to 25 °C) for 15 minutes.
Answer 2: Thank you for your suggestion. We have made corrections accordingly.
Comment 3:
Please correct KPC in this sentence : The genes blaNDM, blaKP, blaOXA-48, and blaVIM were detected by multiplex PCR reaction’.
Answer 3: Thank you for pointing out our typing mistake. We have corrected it.
Comment 4:
Write ‘The PCR conditions were: initial denaturation at 94 °C for 5 min, followed
by 30 cycles of denaturation at 94 °C for 45 s, annealing at 59 °C for 60 s, and extension at 72 °C for 60 s, and the final extension at 72 °C for 10 min.’
Answer 4: As per your suggestion, we have made corrections.
Comment 5:
Please correct ‘blaNDM Rw’ an blaOXA-48-like in table 1.
Answer 5: We have corrected the mistake in Table 1.
Comment 6:
You can delete the footnote under table1; really it is not necessary it exist in the manuscript and are well known enzymes (KPC – Klebsiella pneumoniae carbapenemase; VIM – Verona integron-encoded metallo--lactamase; NDM – New Delhi metallo-β-lactamase; IMP – imipenemase; OXA-48-like – Oxacillinase-48-like).
Answer 6: We have followed the suggestion and deleted the footnote under Table 1.
Comment 7:
Table 2, you write frequency, this is not a frequency this is a number. So I advide to write simply: Number of isolates (Percentage %), so it is a unique column, for example for the line urine culture in its front just one column containing: 34 (1.9), for PCR OXA-48 /Negative/ 102 (32.7), ......the remaining lines also........... I hope that you understand me.
Answer 7: Yes, we understood you, and we are very grateful for your suggestions and advice. We have redesigned Table 2 accordingly.
Comment 8:
Please below table 3, add abbreviation of TP, FP, TN, FN
Answer 8: Thank you for the suggestion. We have corrected this oversight and added abbreviations of TP, FP, TN, and FN.
Comment 9:
In this sentence ‘The overall agreement between NG-Test Carba 5 and 145
PCR across all gene targets was moderate (κ = 0.495, p < 0.001) (Table 3).’ I believe you mean table 4 not table 3.
Answer 9: You are correct. Thank you for bringing this to our attention. We mistakenly wrote Table 3 instead of Table 4, but we have corrected it.
Comment 10:
You wrote ‘There are several possible explanations for this. Firstly we speculated on possibility that bacterial colonies utilized to prepare immunochromatographic tests and DNA isolation may contain clones that were not recognized, and which could influence our data analysis’. THIS hypothesis is not clear, as written it is not correct scientifically, I advise that you read this and see if it correspond really to what you want say. Clones not recognized?!
Please verify this hypothesis it seems not correct or not clear ‘Finally, a PCR may yield false-positive results if a mutation alters the carbapenemase activity but not the primer binding sites or gene (40).’ You mean that the gene exists but not expressed owing to mutation(s) in the structural gene or in its promoter.
Answer 10:
Thank you for your comment, we changed this sentence:
There are several possible explanations for this. Firstly we speculated on possibility that bacterial colonies utilized to prepare immunochromatographic tests and DNA isolation may contain clones that were not recognized, and which could influence our data analysis’
to be more clear as follows:
There are several possible explanations for this. Firstly we speculated on possibility that bacterial colonies utilized to prepare immunochromatographic tests and DNA isolation may contain clonally related isolates with different resistotypes, that were not recognized, and which could influence our data analysis
Thank you for your comment. We have changed this sentence as suggested, so it would be clearer as follows:
Finally, a PCR may yield false-positive results due to mutations in the structural gene or in its promoter, which impair its expression. (45) Also, the reference is changed it is not (40) but (45) because we made other correction in the manuscript.
Reviewer 4 Report
Comments and Suggestions for Authors
Pls find attached

Needs to be improved
Author Response
Evaluation of The Ng-Test Carba 5 for Rapid Detection of Carbapenemases in Clinical Isolates of Klebsiella pneumoniae
Title
Comment 1:
- Suggest to change title as “Evaluation of the Ng-Test Carba 5 for Rapid Detection of Carbapenemases in Klebsiella pneumoniae”. Follow standard guidelines to write name of the organism.
Answer 1: Thank you for noticing and bringing to our attention the mistake in the title. We have corrected it accordingly.
Introduction
Comment 1:
- Line no 31- Grammatically incorrect sentence. Please re-write.
Answer 1: We have rewritten Introduction section according to Comment 3.
Comment 2:
- Line 43- Remove full stop after “”
Answer 2: Thank you for noticing this mistake. We have corrected it.
Comment 3:
- The introduction is written with incorrect sentences, and without a proper flow. Without just jumping into CRKp, first write about the burden of antibiotic resistance and carbapenem resistance. Then mention about pneumoniae and why carbapenem resistance is increasingly emerging among K. pneumoniae isolates. Mention the burden and prevalence of CRKp globally in your country/study setting.
- Answer 3: As per your suggestion we have rewritten Introduction section to make it more coherent. We also mentioned the prevalence of CRKp.
Methodology
Comment 1:
- Suggest mentioning the types of clinical samples included in this study.
Answer 1: According to your suggestion, we have added types of clinical samples in our study.
Comment 2:
- Line 84- The word “ETEST” should be correctly written.
Answer 2: Thank you for your suggestion. We have corrected accordingly. We have changed ’E TEST’ to ’quantitative assay for determining MIC of antimicrobial agents, MIC Test Strip (Liofilchem, Italy)’.
Comment 3:
- The authors have mentioned “According to EUCAST guidelines, pneumoniae isolates that demonstrated an inhibition zone for ertapenem or/and meropenem ˂25 mm using the disc diffusion method and < 2 mg/L by gradient test method were chosen for additional examination.” Here, the reason for selecting this cutoff/ interpretation criteria should be mentioned.
- Answer 3: We used the EUCAST guidelines related with carbapenem-resistance of pneumoniae as the first criteria for isolates inclusion in our study.
Comment 4:
- In 2.2, the methodology of “NG-Test Carba 5 Assay” is not well explained. It should be explained clearly, to give a clear understanding to the reader. The methodology should be informative enough to the readers to reproduce.
Answer 4: As per your suggestion, we have added detailed methodology for “NG-Test Carba 5 Assay”
Comment 5:
- Remove the word in bold in line 99.
Answer 5: We have corrected according to the suggestion.
Comment 6:
- In line 106, correct as “blaKPC” and “blaKPC”.
Answer 6: We have corrected according to the suggestion.
Comment 7:
- Remove the additional space before oC throughout the manuscript.
Answer 7: We have made appropriate corrections.
Comment 8:
- Several spelling mistakes and formatting errors were found. Eg: in line 109-prducts, in line 113-Verona integron-encoded metallo--lactamase. Suggest to check throughout the manuscript and correct.
Answer 8: Thank you for pointing out our typing mistakes. We once again reviewed entire manuscript and corrected every mistake we encountered.
Results
Comment 1:
- The table 4 is not cited in the text.
Answer 1: Thank you for your comment. We have made changes to this section and cited Table 4 appropriately.
Comment 2:
- Tables should be understandable itself without referring to the text. Suggest adding footers to each tale and describe abbreviations. Also, caption of Table 2 is lacks clarity.
Answer 2: According to suggestion, we have made appropriate corrections to make the tables more understandable.
Comment 3:
- Suggest dividing separate results sections described in different tables in to separate paragraphs rather than presenting as a single paragraph.
Answer 3: Thank you for your suggestion. We have divided the Results into three sections coressponding to each table.
Discussion
Comment 1:
- Repetitions were found in the discussion section. Eg: when describing test accuracies.
Answer 1: We are grateful for this suggestion. We have made some changes in this section in order to avoid unnecessary repetition.
Comment 2:
- Although high prevalence of OXA-48 is mentioned, its significance in local epidemiological context is not mentioned explicitly. Also, the absence of IMP and low detection of VIM should be discussed beyond test performance, with reported regional gene circulation patterns.
Answer 2: Although the focus of our study is not on the carbapenemase prevalence and epidemiological context, as per your suggestion, we reported in one paragraph carbapenemase genes circulation in our country.
Comment 3:
- The results showed high individual sensitivities/ specificities but only moderate overall kappa. This finding is not discussed.
- Answer 3: Thank you for your valuable comment. We are aware that the overall kappa was only moderate despite the results showing high individual sensitivity/specificity, and we have presented these results in Table 4. This difference is primarily due to the lower test performance for VIM detection, which affected the overall coefficient of agreement, as we point out in the discussion section of the paper.
Comment 4:
- Although, treatment options are mentioned, direct implications of co-occurring multiple carbapenemase genes for therapy and infection control are not discussed
Answer 4: Answer: Thank you for your valuable comment. We agree that the co-occurring multiple carbapenemase genes can have important therapeutic implications. However, the primary aim of our study was to evaluate the performance of NG-Test CARBA 5, with which in our routine work we help the clinician to correctly prescribe therapy
Comment 5:
- Limitations – The authors have not discussed the limitations that occurred due to inclusion of a single study setting, and limited sample diversity
Answer 5: Thank you for your comments, however our aim was to estimet test performance and to incorporate it in routin workflow in our hospital/settings. Also, we think that (urine culture, blood culture, bronchoalveolar lavage (BAL), aspirate, kiss swab, and liquor) are the most frequent samples in our laboratory, so we were encourage to conduct a test assessment based on these sample types.
Comment 6:
- Follow the journal guidelines. It is not recommended to write some words in bold font in the text.
Answer 6: Thank you for pointing out this mistake. We have corrected this section.
Conclusion
Comment 1:
- Suggest improving conclusion incorporating your findings.
Answer 1: Thank you for suggestion. We have added a sentence that summerizes our main finding.
References
Comment 1:
- References are not formatted according to journal requirements. Eg: organisms’ names are not in italics in references, some references are underlined etc.
Answer 1: Thank you for your comment. We sent our manuscript without formating references according to journal requirements since journal accepts free format submission at this step. However, now we have formated references according to journal requirements.
Comment 2
- The manuscript lacks, author contributions, funding, informed consent, data availability, conflicts of interest, abbreviations sections which are mentioned in submission guide. Suggest adding those.
Answer 2: We have provided all these documents durring initial submission to the journal.
Comment 3
- Also, the journal suggests adding the “methods” section at the latter part of the manuscript.
Answer 3: Thank you for your comment. As in the case of references, we sent our manuscript without formating according to journal requirements since journal accepts free format submission at initial step.
Round 2
Reviewer 4 Report
Comments and Suggestions for Authors
The article can be accepted in its current format,Thank you for taking time to do the changes